# Prevalence and associated factors of gender-based violence for female: Evidence from school students in Nepal-A cross-sectional study

**Laxmi Gautam** [1]*, **Manisha Shah** [1], **Durga Khadka Mishra**[2], **Padam Kanta Dahal**[3], **Sujan Gautam** [1,4]

**1** Department of Public Health, Manmohan Memorial Institute of Health Sciences, Kathmandu, Nepal,
**2** Madan Bhandari Academy of Health Sciences, Hetauda, Nepal, **3** School of Health, Medical and Applied Sciences, Central Queensland University, Sydney Campus, Queensland, NSW, Australia, **4** The Institute of Fundamental Research and Studies (InFeRS), Kathmandu, Nepal

* laxmi.dp26@gmail.com

## Abstract

Gender-based violence (GBV) is a major global public health challenge in the 21st century that poses a serious impact on women's health and well-being. Therefore, this study aimed to assess the prevalence and factors associated with GBV among secondary school female students in the Sarlahi district of Nepal. Using a cross-sectional study, we collected data from 225 secondary-level female students in the Sarlahi district of Nepal. Data was collected by using a semi-structured, self-administered questionnaire. Probability proportionate and simple random sampling techniques were used for sampling. The association was explored by using a chi-square test and binary logistic regression. The two-tailed significance level for all analyses was set at p<0.05. The overall prevalence of GBV among the students during their lifetime was 45.33 of which physical violence was 16.89%, sexual violence was 30.22% and psychological violence was 39.56%. The prevalence of experiencing physical violence from family members was 97.36%, followed by emotional violence (41.57%). Further, the prevalence of sexual violence from non-family members was 91.17%. Type of family had a significant association with lifetime experience of GBV (p = 0.003). Gender based discrimination in the family had a significant association with lifetime (p = 0.001) as well as last 12 months (p = 0.001) GBV experience. Experience of witnessing physical violence as a child within the last 12 months was associated with GBV (p = 0.03). Different forms of GBV such as physical, sexual, and emotional acts of violence among female students were highly prevalent. However, their knowledge and awareness of confronting this issue were limited. This warrants the urgent need to establish preventive and responsive control measures within schools and communities to address GBV effectively.

**Data Availability Statement:** All relevant data generated in the study is included in the paper and its Supporting Information files.

**Funding:** The authors received no specific funding for this work.

**Competing interests:** The authors have declared that no competing interests exist.

## Introduction

The United Nations has defined GBV as, "any act of gender-based violence that results in, or is likely to result in, physical, sexual, or mental harm or suffering to women, including threats of such acts, coercion or arbitrary deprivation of liberty, whether occurring in public or in private life" [1]. Gender-based violence (GBV) remains one of the most serious social, legal, and health challenges of the 21st century. It is a major public health problem and human rights concern throughout the world that has a serious impact on women's health and well-being [2]. GBV or violence against women and girls (VAWG), is a global pandemic that affects 1 in 3 women in their lifetime and 35% of women worldwide have experienced either physical and/or sexual intimate partner violence or non-partner sexual violence [3]. Further, World Health Organization (WHO) estimated that 1 in 5 women between 15–24 years old could be subject to GBV from their intimate partners, while 40.2% of all women experience physical and/or sexual intimate partner or non-partner violence in their lifetime in southeast Asia [4]. Within the South-Asian region, the magnitude of GBV varied significantly. The highest proportion (79.6%) was reported from India [5] followed by Pakistan (38.40) [6] and Maldives (34.6%) [7]. In Nepal, the prevalence of GBV is rapidly increased over few decades which has created a serious impact on the health and well-being of females. Evidence shows that over 150 females were killed as a result of GBV in 2017, highlighted the major perpetrator was family member or relatives [8]. Further, another study conducted among community people in Kathmandu, Lalitpur, Bhaktapur, and Nuwakot districts in 2018 showed 11.3% of respondents experienced GBV in past 12 months of survey [9]. Nepal Demographic Health Survey in 2022 showed that 23% of women aged 15–49 have experienced physical violence since age of 15 and 8%, which is a percentage higher compared to 2016 estimates [10,11].

Violence against women and girls is a human rights violation and can have devastating health consequences including death. GBV has significant and long-lasting impacts on physical and mental health and injury, unintended pregnancy and pregnancy complications, sexually transmitted infections, HIV, depression, post-traumatic stress disorder, and even death [12]. Additionally, violence negatively affects women's general well-being and prevents women from fully participating in society [13]. It impacts their families, their community, and the country at large. GBV occurs as a result of the normative role expectations associated with each gender and unequal power relationships within the context of a specific society. Domestic violence, marital rape, dowry-related violence, child marriage, polygamy, female infanticide, witchcraft accusations, chhaupadi, and trafficking of women and girls for sexual exploitation are common types of GBV in Nepal [14]. Furthermore, patriarchal attitudes and deep-rooted stereotypes that discriminate against women remain entrenched in social, cultural, religious, socio-economic and political institutions. However, studies are lacking to estimate the prevalence of gender-based violence at the school level in low and middle-income countries like Nepal and insufficient to design a responsive control intervention to fight against the problem.

Furthermore, patriarchal attitudes and deep-rooted stereotypes that discriminate against women remain entrenched in social, cultural, religious, socio-economic and political institutions. Despite the growing concern about violence against women, there is limited understanding regarding the underlying risk factors of GBV related to middle and late adolescence in low and middle-income countries like Nepal. This lack of evidence hinders the design and implementation of the responsive control intervention to fight against the problem. This concern was the basis for the conception of this study. Therefore, this study aimed to assess the prevalence and underlying factors associated with gender-based violence among female secondary-level students.

## Materials and methods

### Study design and population

A cross-sectional study was conducted among female secondary—level students in grade 9–12 in Brahmpuri Rural Municipality of Sarlahi district, Nepal. The overall time period for the information collection was from 23/10/2020 to 06/01/2021. A sample of 225 participant was estimated using Cochran's formula with reference to prevalence of GBV (Prevalence = 48%) from a study conducted by Government of Nepal [15] at 5% allowable error and 10% non-response rate.The Sarlahi district was selected purposively because percentage of women who were able to make specific decisions was lower in Madhesh Province and was lowest among all 7 provinces of Nepal Madhesh Province [10] and the Sarlahi district is one of the core areas of Madhesh Province.

First of all, the list of schools was obtained from the educational section of the Brahmpuri Rural Municipality. Of the total schools in that rural municipality, only four schools met our criteria (teaching grade 9th to 12th). Out of four schools, two schools were selected using Simple Random Sampling method. Then, the list of female students from grade 9th to 12th was obtained from each school. Probability Proportionate Allocation (PPA) was used to select sample (female students) from selected schools, and were selected randomly using a simple random sampling method. Those students who were absent during the data collection period were excluded from the study. This study was adhered to the Strengthening the Reporting of Observational Studies in Epidemiology (STROBE) guidelines for reporting observational studies.

### Outcome measures

Gender-based violence was the major outcome of this study including the prevalence of physical, sexual, and Psychological violence. In this study, physical violence includes if the respondents reported to have experienced slapping, punching, kicking/dragging, beating/hitting with any object, cutting/ biting, shaking, shoving, pushing, throwing, and burning/chocking against them. Similarly for sexual violence, it was considered if the respondent had experienced unwanted or non-consensual sexual act through force or threat against them. In case of psychological violence, if there was systemic destruction of women's or girl's self-esteem and/or sense of safety, it was considered violence against them, which included humiliation, being made to feel unwanted, forced isolation from family or friends, threats to harm the individual or someone they care about, repeated yelling or degradation, inducing fear through intimidating words or gestures, controlling behavior, and the destruction of possessions. To measure these experiences, different questions were developed based on the reference provided by WHO, UNICEF and UN women and were modified to make applicable in local context of the respondents [14,16–18]. (S1 File) The Prevalence of GBV was measured if the respondents had experienced any type of physical, sexual, psychological violence that targeted them.

### Statistical analyses

The frequency and percentage were reported for categorical variables, and mean and standard deviation for quantitative variable. The association between dependent and independent variables was determined by chi-square test and binary logistic regression. Further, multivariate analyses was conducted to test the association of different independent variables with the experience of violence in "lifetime" and "last 12 months of the survey". A probability (p) value less than 0.05 was considered statistically significant. Microsoft Excel was used for data cleaning and coding and statistical package of social science (SPSS) version 16 was used for statistical analyses (complete data in S1 Data).

## Ethical statement

Ethical approval was received from the institutional review committee (IRC) of the Manmohan Memorial Institute of Health Sciences (MMIHS-IRC 77/13) (Evidence of Ethical approval in S3 File). The study was conducted from October 2020 to January 2021. Further, permission was obtained from each school and municipality before the information collection. Written informed consent was obtained from participants after explaining purpose of the study prior to the data collection. In case of participants below 18 years of age, it was not feasible to reach the parents of each respondents as the resources was constrained. Therefore, permission was obtained from the respective school principals, and class teachers, and assent was also obtained from the participants. Additionally, participants were informed about the privacy and confidentiality of information, as well as the rights to withdraw and refuse the study at any time. All other ethical guidelines were followed throughout the process of this study.

## Inclusivity in global research

Additional information regarding the ethical, cultural, and scientific considerations specific to inclusivity in global research is included in the Supporting Information. (Inclusivity-in-global-research-questionnaire in S2 File).

# Results

The mean age of respondents was 17.17±2.48 years, and more than two third of them were in age group of 15–20 years. Their father and grandfather were the major decision makers in their family. Alcohol and drug abusers in the family were reported by more than one third of the respondents. Most of the respondents needed permission from their families to go to health institution, while more than a quarter (28.4%) faced discrimination in comparison to their male counterparts. Almost 30% of respondents had witnessed GBV as a child, followed by sexual (10.7%) and physical violence (8.9%) (**Table 1**).

Almost half (45.3%) of the respondents had experienced GBV in their lifetime, followed by the last 12 months (35.1%) preceding the survey. The lifetime physical, sexual, and psychological violence among the respondents was 16.89%, 30.22%, and 39.56%, respectively. Regarding the perpetrators of GBV, physical violence was mainly from family members (97%), while sexual violence was mainly from non-family members (91%). Less than half (42.15%) of the respondents had reported to someone after violence because 38.98% had no idea that was a crime and 20.33% did not report due to fear (**Table 2**).

Regarding the lifetime experiences of GBV of any form; it was higher among the younger age group, which was 60.98% among the age group <15 in comparison to 25.0% among the age group >20 years. Similarly, it was more common among non-Hindus by religion than Hindus. GBV was more common in joint and extended families than in nuclear families and was more common if the parents were living separately. The lifetime experience was mostly inversely proportional to the educational status of parents. Likewise, girls were more likely to face GBV if they needed permission from family members to visit health center or hospital, if they discriminated between men and women in family and if they had experienced witnessing violence as a child (**Table 3**).

Regarding the GBV, in the 12 months preceding the survey, age and prevalence of violence had a direct association. Similarly, GBV was experienced more in joint families and was significantly associated with those whose parents were separated, the educational status of parents, the existence of alcohol or drug abusers in the family, discrimination between men and women in the family, and the experience of witnessing GBV as a child in the last 12 months before the survey (**Table 4**)

**Table 1. Descriptive characteristics of the study participants.**

| Variables | Frequency | Percentage |
|---|---|---|
| Age in years | | |
| <15 | 41 | 18.2 |
| 15–20 | 152 | 67.6 |
| >20 | 32 | 14.2 |
| Religion | | |
| Hindu | 189 | 84.0 |
| Other than Hindu | 36 | 16.0 |
| Type of Family | | |
| Nuclear family | 126 | 56.0 |
| Joint family and extended | 99 | 44.0 |
| Living status of Parents | | |
| Together | 183 | 81.3 |
| Separately | 42 | 18.7 |
| Educational status of father (n = 222) | | |
| Higher Education (bachelors and above) | 10 | 4.5 |
| Secondary Education (9–12) | 75 | 33.8 |
| Primary Education (1–8) | 85 | 38.3 |
| No formal Education | 28 | 12.6 |
| Illiterate | 24 | 10.8 |
| Educational status of Mother (n = 219) | | |
| Secondary Education (9–12) | 16 | 7.3 |
| Primary Education (1–8) | 75 | 34.2 |
| No formal Education | 58 | 26.5 |
| Illiterate | 70 | 32.0 |
| Decision maker of the family | | |
| Father/Grandfather | 201 | 89.3 |
| Mother/Grandmother | 15 | 6.7 |
| Others | 9 | 4.0 |
| Alcohol or drugs abuser in the family | | |
| No | 138 | 61.3 |
| Yes | 87 | 38.7 |
| Need permission from family members to visit health center or hospital | | |
| No | 4 | 1.8 |
| Yes | 221 | 98.2 |
| Discriminate between men and women in family or treats them differently | | |
| No | 161 | 71.6 |
| Yes | 64 | 28.4 |
| Experience of Witnessing GBV as a child (n = 218) | | |
| No | 155 | 71.1 |
| Yes | 63 | 28.9 |
| Experience of witnessing physical violence as a child | | |
| No | 205 | 91.1 |
| Yes | 20 | 8.9 |
| Experience of witnessing sexual violence as a child | | |
| No | 201 | 89.3 |
| Yes | 24 | 10.7 |

**Table 2. Prevalence of different forms of GBV, perpetrators, and reporting status after violence.**

| Variables | Frequency | Percentage (%) |
|---|---|---|
| **Ever Experienced GBV in lifetime** | | |
| Yes | 102 | 45.3 |
| No | 123 | 54.7 |
| **Experienced GBV in last 12 months preceding study** | | |
| Yes | 79 | 35.1 |
| No | 146 | 64.9 |
| **Physical violence** | | |
| Yes | 38 | 16.89 |
| No | 187 | 83.11 |
| **Physical Violence in last 12 months (n = 38)** | | |
| Yes | 14 | 36.8 |
| No | 24 | 63.2 |
| **Sexual violence** | | |
| Yes | 68 | 30.22 |
| No | 157 | 69.78 |
| **Sexual Violence In last 12 months (n = 49)** | | |
| Yes | 39 | 57 |
| No | 29 | 43 |
| **Emotional violence** | | |
| Yes | 90 | 40 |
| No | 135 | 60 |
| **Emotional Violence in last 12 months (n = 90)** | | |
| Yes | 60 | 66.67 |
| No | 30 | 33.33 |
| **Perpetrators** | | |
| **Physical violence** | | |
| Family member | 37 | 97 |
| Non-Family members | 1 | 3 |
| **Sexual violence** | | |
| Family member | 6 | 9 |
| Non-Family members | 62 | 91 |
| **Emotional violence** | | |
| Family member | 37 | 42 |
| Non-Family members | 53 | 58 |
| **Reporting after Violence (n = 102)** | | |
| Yes | 43 | 42.15 |
| No | 59 | 57.84 |
| **Reasons for not reporting violence (n = 59)** | | |
| No idea that was crime | 23 | 38.98 |
| Scared to report | 12 | 20.33 |
| Because of family prestige | 11 | 18.64 |
| Embarrassment | 7 | 11.86 |
| Accept violence as normal | 6 | 10.16 |

GBV in lifetime was 4.68 (1.69–12.95) times more likely to happen among respondents aged more than 20 compared to the < 15 years age group. Similarly, there were 4.56 (2.03–10.23) times more likely to face GBV by non-Hindus by religion in comparison to Hindu

**Table 3. Ever experience of GBV by background characteristics.**

| Variables | Experienced GBV ever | | P value |
|---|---|---|---|
| Age in years | Yes (%) | No (%) | |
| <15 | 25 (60.98) | 16 (39.02) | 0.001 |
| 15–20 | 90 (59.2) | 62 (40.8) | |
| >20 | 8 (25.0) | 24 (75.0) | |
| Religion | | | |
| Hindu | 75 (39.7) | 114 (60.3) | <0.001 |
| Other than Hindu | 27 (75.0) | 9 (25.0) | |
| Type of Family | | | |
| Nuclear family | 46 (36.5) | 8 0 (63.5) | 0.003 |
| Joint family and extended | 56 (56.6) | 43 (43.4) | |
| Living status of Parents | | | |
| Together | 65 (35.5) | 118 (64.5) | <0.001 |
| Separately | 37 (88.1) | 5 (11.9) | |
| Educational status of father (n = 222) | | | |
| Higher Education (bachelors and above) | 1 (10.0) | 9 (90.0) | <0.001 |
| Secondary Education (9–12) | 14 (18.7) | 61 (81.3) | |
| Primary Education (1–8) | 47(55.3) | 38 (44.7) | |
| No Formal Education | 17 (60.7) | 11 (39.3) | |
| Illiterate | 20 (83.3) | 4 (16.7) | |
| Educational status of Mother (n = 219) | | | |
| Secondary Education (9–12) | 5 (31.3) | 11 (68.8) | <0.001 |
| Primary Education (1–8) | 14 (18.7) | 61 (81.3) | |
| No Formal Education | 28 (48.3) | 30 (51.7) | |
| Illiterate | 49 (70.0) | 21 (30.3) | |
| Decision maker of the family | | | |
| Father/Grandfather | 83 (41.3) | 118 (58.7) | 0.002 |
| Mother/Grandmother | 11 (73.3) | 4 (26.7) | |
| Others | 8 (88.9) | 1 (11.1) | |
| Alcohol or drugs abuser in the family | | | |
| No | 36 (26.1) | 102 (73.9) | <0.001 |
| Yes | 66 (75.9) | 21 (24.1) | |
| Need permission from family members to visit health center or hospital | | | |
| No | 0 (0) | 4 (100) | 0.066 |
| Yes | 102 (46.2) | 119 (53.8) | |
| Discriminate between men and women in family or treats them differently | | | |
| No | 48 (29.8) | 113 (70.2) | <0.001 |
| Yes | 54 (84.4) | 10 (15.6) | |
| Experience of Witnessing GBV as a child (n = 218) | | | |
| No | 51 (32.9) | 104 (67.1) | <0.001 |
| Yes | 47 (74.6) | 16 (25.4) | |
| Experience of witnessing physical violence as a child | | | |
| No | 84 (40.8) | 122 (59.2) | <0.001 |
| Yes | 18 (94.7) | 1 (5.3) | |
| Experience of witnessing sexual violence as a child | | | |
| No | 79 (39.3) | 122 (60.7) | <0.001 |
| Yes | 23 (95.8) | 1 (4.2) | |

**Table 4. Experience of GBV in the year preceding the survey by background characteristics.**

| Background variables | GBV in the last 12 months | | P value |
|---|---|---|---|
| | Yes (%) | No (%) | |
| **Age** | | | |
| **<15** | 10 (24.4) | 31 (75.6) | 0.005 |
| **15–20** | 50 (32.9) | 102 (67.1) | |
| **>20** | 19 (59.4) | 13 (40.6) | |
| **Religion** | | | |
| **Hindu** | 59 (31.2) | 130 (68.8) | 0.005 |
| **Others than Hindu** | 20 (55.6) | 16 (44.4) | |
| **Type of family** | | | |
| **Nuclear family** | 32 (25.4) | 94 (74.6) | 0.001 |
| **Joint family and extended** | 47 (47.5) | 52 (52.5) | |
| **Living status of Parents** | | | |
| **Together** | 49 (26.8) | 134 (73.2) | <0.001 |
| **Separately** | 30 (71.4) | 12 (28.6) | |
| **Education of Father++ (n = 222)** | | | |
| **Illiterate** | 16 (66.7) | 8 (33.3) | <0.001 |
| **No formal Education** | 15 (53.6) | 13 (46.4) | |
| **Primary Education (1–8)** | 39 (45.9) | 46 (54.1) | |
| **Secondary Education (9–12)** | 8 (10.7) | 67 (89.3) | |
| **Higher Education (bachelors and above)** | 0 (0) | 10 (100) | |
| **Education of mother (n = 219)** | | | |
| **Secondary Education (9–12)** | 1 (6.3) | 15 (93.8) | <0.001 |
| **Primary Education (1–8)** | 11 (14.7) | 64 (85.3) | |
| No formal Education | 23 (39.7) | 35 (60.3) | |
| **Illiterate** | 39 (55.7) | 31 (44.3) | |
| **Decision maker in the family** | | | |
| **Father/Grandfather** | 64 (31.8) | 137 (68.2) | 0.006 |
| **Mother/Grandmother** | 8 (53.3) | 7 (46.7) | |
| **Others** | 7 (77.8) | 2 (22.2) | |
| **Alcohol or drugs abuser in the family** | | | |
| **No** | 55 (63.2) | 32 (36.8) | <0.001 |
| **Yes** | 24 (17.4) | 114 (82.6) | |
| **Need permission from family members to visit health center or hospital** | | | |
| **No** | 0 (0.0) | 4 (100) | 0.138 |
| **Yes** | 79 (35.7) | 142 (64.3) | |
| **Discriminate between men and women in family or treat them differently** | | | |
| **No** | 34 (21.1) | 127 (78.9) | <0.001 |
| **Yes** | 45 (70.3) | 19 (29.7) | |
| **Experience of Witnessing GBV as a child (n = 218)** | | | |
| **No** | 39 (61.9) | 24 (38.1) | <0.001 |
| **Yes** | 37 (23.9) | 118 (76.1) | |
| **Experience of witnessing physical violence as a child** | | | |
| **No** | 15 (78.9) | 4 (21.1) | <0.001 |
| **Yes** | 64 (31.1) | 142 (68.9) | |
| **Experience of witnessing sexual violence as a child** | | | |
| **No** | 18 (75.0) | 6 (25.0) | <0.001 |

*(Continued)*

**Table 4.** (Continued)

| Background variables | GBV in the last 12 months | | P value |
|---|---|---|---|
| | Yes (%) | No (%) | |
| **Yes** | 61 (30.3) | 140 (69.7) | |

++: Cannot be computed because of zero value in one of the cells (None of the students reported having experienced GBV in the last 12 months whose father had university level of education).

respondents. Respondents whose parents were not living together were more likely to face GBV 13.43 (5.03–35.85) times than those whose parents were living together. The higher the education of parents, the lesser the chances of facing GBV. The alcohol or drug abuser in the family also contributed to the increasing likelihood of GBV by 8.91 (4.78–16.57) times and discrimination in the family among men and women by 12.71 (5.97–27.03) times. Regarding the childhood experience of witnessing violence, those who witnessed GBV were 5.99 (3.01–11.57) times more at odd of facing GBV later. In the case of GBV 12 months preceding the survey, the odds of facing GBV were increasing along with age, more among religions other than Hindu and more among joint families compared to nuclear families. Those respondents whose parents were not together were more at risk of facing GBV (6.84 (3.24–14.40)) than those whose parents were living together. Alcohol or drug abusers in families had increased the odds of GBV by 8.16 (4.39–15.16) times; similarly, those respondents who faced discrimination in family were 8.08 (4.58–17.05) times more likely to face GBV compared to those who didn't face discrimination within 12 months prior to the survey. Type of family had a significant association with lifetime experience of GBV (p = 0.02) while the living status of parents was associated with 12 months experience of GBV (p = 0.02) and decision maker in the family was associated with both. Discrimination between men and women in families had a significant association with lifetime and 12 months GBV experience (p = 0.01 and p = 0.001) (**Table 5**).

## Discussion

GBV is widely recognized as a major public health problem and is a human rights violation. It not only affects the health and dignity of the victims but also their families and generations to come. Many studies have been conducted among women of reproductive age to estimate the prevalence and establish evidence for effective interventions that address GBV [2,19,20]. GBV against women and girls is a product of interactions between familial and social relationships within an environment. Although there are numerous similar studies conducted among women of reproductive age, very few studies are conducted focusing particularly among adolescents girls. Therefore, to address this gap, this study was conducted which aims to explore the prevalence and factors of GBV among the secondary school female students in Nepal.

This study revealed that the risk of GBV like physical, mental, and sexual violence among secondary-level female students is still high. Contributing factors like joint family status, poor education status of parents, single parents, and the existence of alcohol or drug abusers in the family had a significant association with the violence faced by the study population. Further, GBV among the teenage girls is highly prevalent in both the lifetime and 12 months prior to the survey. This study identified a higher risk of GBV among the girls who were studying at higher secondary level, which is supported by the gender gap index of 0.659 in 2023 [21]. In line with this study, previous studies conducted in Debre Markos Town [22], Nepal [15] and Ethiopia [23] showed that the prevalence of GBV among girls was 47%, 48%, and 47.2%

**Table 5. Multi Factors associated with experiencing GBV "Ever" and "within 12 months" among participants.**

| Variables | Experiencing GBV ever | | | | Experiencing GBV 12 months | | | |
|---|---|---|---|---|---|---|---|---|
| | COR (95% CI) | p-value | AOR (95% CI) | p-value | COR (95% CI) | p-value | AOR (95% CI) | p-value |
| **Age** | | | | | | | | |
| **<15** | Ref | | | | Ref | | | |
| **15–20** | 1.07 (0.53–2.18) | 0.838 | 0.60 (0.20–1.78) | 0.36 | 1.52 (0.69–3.34) | 0.299 | 1.60 (0.49–5.26) | 0.43 |
| **>20** | 4.68 (1.69–12.95) | 0.003 | 0.52 (0.08–3.24) | 0.49 | 4.53 (1.66–12.35) | 0.003 | 1.33 (0.26–6.72) | 0.73 |
| **Religion** | | | | | | | | |
| **Hindu** | Ref | | Ref | | Ref | | Ref | |
| **Others than Hindu** | 4.56 (2.03–10.23) | 0.005 | 3.41 (0.69–16.94) | 0.13 | 2.75 (1.33–5.69) | 0.006 | 0.82 (0.21–3.26) | 0.78 |
| **Type of family** | | | | | | | | |
| **Nuclear family** | Ref | | Ref | | Ref | | Ref | |
| **Joint family and extended** | 2.26 (1.32–3.87) | 0.003 | 0.28 (0.09–0.82) | 0.02 | 2.65 (1.51–4.66) | 0.001 | 0.43 (0.15–1.26) | 0.12 |
| **Living status of Parents** | | | | | | | | |
| **Together** | Ref | | Ref | | Ref | | Ref | |
| **Separately** | 13.43 (5.03–35.85) | <0.001 | 8.493(2.093–34.456) | 0.003 | 6.84 (3.24–14.40) | <0.001 | 4.45 (1.48–13.99) | 0.008 |
| **Education of Father++ (n = 222) NA NA NA NA** | | | | | NA | NA | NA | NA |
| **Education of mother (n = 219)** | | | | | | | | |
| **Secondary Education (9–12)** | Ref | | Ref | | Ref | | Ref | |
| **Primary Education (1–8)** | 0.505 (0.15–1.68) | 0.230 | 0.30 (0.03–2.47) | 0.81 | 2.578 (0.30–21.54) | 0.032 | 1.10 (0.07–17.18) | 0.946 |
| No formal Education | 2.053 (0.63–6.65) | 0.006 | 0.75 (0.07–7.26) | 0.71 | 9.857 (1.21–79.81) | 0.006 | 3.59 (0.24–53.37) | 0.352 |
| Illiterate | 5.133 (1.58–16.61) | <0.001 | 1.60 (0.13–18.36) | 0.16 | 18.87 (2.36–150.81) | <0.001 | 6.32 (0.40–99.75) | 0.190 |
| **Decision maker in the family** | | | | | | | | |
| **Father/Grandfather** | Ref | | Ref | | Ref | | Ref | |
| **Mother/Grandmother** | 3.910 (1.20–12.70) | 0.023 | 17.36 (2.95–102.13) | 0.00 | 2.24 (0.85–7.03) | 0.097 | 9.17 (1.76–47.58) | 0.008 |
| **Others** | 11.373 (1.39–92.66) | 0.023 | 9.10 (0.33–249.96) | 0.19 | 7.49 (1.51–37.08) | 0.14 | 17.33 (1.14–268.41) | 0.040 |
| **Alcohol or drugs abuser in the family** | | | | | | | | |
| **No** | Ref | | Ref | | Ref | | Ref | |
| **Yes** | 8.91 (4.78–16.57) | <0.001 | 1.88 (0.63–5.64) | 0.26 | 8.16 (4.39–15.16) | <0.001 | 1.88 (0.65–5.40) | 0.24 |
| **Discriminate between men and women in family or treats them differently** | | | | | | | | |
| **No** | Ref | | Ref | | Ref | | Ref | |
| **Yes** | 12.71 (5.97–27.03) | <0.001 | 4.83 (1.48–15.66) | 0.01 | 8084 (4.58–17.05) | <0.001 | 6.65 (2.11–20.98) | 0.001 |
| **Experience of Witnessing GBV as a child (n = 218)** | | | | | | | | |
| **No** | Ref | | Ref | | Ref | | Ref | |
| **Yes** | 5.99 (3.01–11.57) | <0.001 | 1.38 (0.43–4.40) | 0.58 | 5.18 (2.76–9.71) | <0.001 | 1.42 (0.48–4.20) | 0.52 |
| **Experience of witnessing physical violence as a child** | | | | | | | | |
| **No** | Ref | | Ref | | Ref | | Ref | |
| **Yes** | 26.14 (3.42–199.60) | 0.002 | 15.29 (1.04–223.51) | 0.05 | 8.32 (2.65–26.06) | <0.001 | 10.11 (1.24–82.16) | 0.03 |
| **Experience of witnessing sexual violence as a child** | | | | | | | | |
| **No** | Ref | | Ref | | Ref | | Ref | |
| **Yes** | 35.51 (4.70–268.29) | 0.001 | 6.73 (0.66–68.35) | 0.11 | 6.68 (2.60–18.19) | 0.00 | 1.007 (0.20–4.84) | 0.99 |

Crude and adjusted odds ratios and 95% confidence intervals from logistic regression analysis.

++: Fathers education cannot be computed because of zero value in one of the cells (None of the students reported having experienced GBV in the last 12 months whose father had university level of education).

respectively. However, our estimates are relatively lower than the study conducted in Southeast Ethiopia (68.2%) and Northwest Ethiopia (67.7%.) [24,25]. These variations might be due to the difference in the status of girls and the gender gap. It warrants not only developing policies but also implementing these policies effectively. Women and girls friendly environments in households as well as educational institutions should be established to make the young generation free from GBV. Also, interventions focused on educating the academic and research community, parents, and students about this important issue can foster respect and equality for girls and women, and create a safe environment.

Similar to this study, previous studies conducted in Ethiopia [23,26] and South Africa [27] showed that experiences of physical violence among the students were 36.2%, 37.99%, and 36.3%, respectively. Further, the Nepal Demographic Health Survey (NDHS) 2022 also prevailed that 23% of women aged 15 to 49 have experienced physical violence since the age of 15 years, including 11% who experienced physical violence [10]. This survey also highlighted that 37% of women of reproductive age experienced physical violence in Madhesh Province, where this study was conducted. In addition, physical violence in both lifetime and 12 months prior to our study is comparable to the study conducted in Kenya, which was 18.0 and 10.7%, respectively [20]. The risk of sexual violence is similar to the previous study conducted in Ethiopia (27.99%) [23]. In contrast, a study in South Africa showed a 46% prevalence of sexual violence [27]. The higher prevalence of sexual violence in these studies could be due to the different study population, which were university students. University students are more mature than school students when it comes to starting intimate relationships and have a higher risk of sexual violence because of the initiation of intimate relationships [28]. Furthermore, NDHS 2022 showed that 8% of women aged 15 to 49 have ever experienced sexual violence, and 4% experienced sexual violence in the 12 months preceding the survey, which was 11% in Madhesh province [10]. The experience of emotional violence in our current study is lower by 50% compared to the previous studies in Ethiopia and South Africa [26,27]. However, NDHS reported that the proportion of women who reported being forced to get married is highest in Madhesh Province (20%), not cared during illness (11.5%), threatened for divorce (10%) were reported by women of reproductive age which might be the drivers of the emotional violence [10].

The present study found that those respondents who faced discrimination between men and women in their families were more at risk of GBV which was supported by a study conducted in Nepal showed that a power play between men and women reinforces inequality and increases the likelihood of violence for women [29]. Further, a study conducted in Sub Saharan Africa showed that having an illiterate mother [OR 2.13] was associated with violence, which is similar to the current study [6]. Similarly, respondents whose father had a university level of education did not face GBV in the last 12 months and the risk of GBV was inversely proportional to the education of the father. This finding was supported by a study in Sub-Sharan Africa; having a father who had completed primary or lower school was three or four times more likely to experience [OR 3.06 and 4.69] sexual violence, respectively, than compared to the respondents who had a father with higher studies [30]. This might be due to mothers with low educational status may be less aware of GBV and unable to teach their daughters about this issue and protect them while they accept violence as a normal process themselves. There are not many studies conducted to examine childhood exposure to violence and its factors in the Nepalese context. However, one of the largest investigations ever conducted on childhood trauma, the adverse childhood experiences (ACE) study in the US, showed that experience of childhood trauma, separation of parents, substance abuse, and experience of domestic violence are the risk factors for GBV among children, which is in line with the findings of the present study [31]. Similarly, students who witnessed physical violence during childhood were (OR:

15.29, CI: 1.04–223.51) more likely to feel violence in their lifetime and also had a significant association with GBV within the last 12 months before the survey (p = 0.03). In the case of the experience of witnessing childhood sexual violence, the risk of GBV was 6.73 (0.66–68.35) times higher than that of those who did not witness such violence in childhood [31]. A systematic review and meta-analysis in education institution of Sub-Sharan Africa shows that in three studies, researchers found witnessing parental violence [OR 2.40, 2.20, and 1.54] was associated with an increased risk of GBV [30].

This study showed that children who were exposed to witnessing violence were 1.38 times more likely to experience GBV in comparison to those who were not exposed to GBV. This finding was supported by the study conducted by Temesgen in Ethiopia which showed that female students who had witnessed parental violence were 1.92 times more likely to experience GBV as compared to those who didn't witness any paternal violence [32]. Substance abuse seems to be a risk factor for GBV in this study, with risk of 1.88 times among the respondents from families with abusers, which was supported by a study conducted in Ethiopia and Sub-Saharan African [30,31]. This may be due to alcohol influencing decision making, and people losing control among themselves, which may lead to GBV. Our study showed that the offenders of physical and sexual violence were mostly family members, which was in line with the findings of the previous study, which showed that family members as offenders of physical violence were 50.2%, compared to 21.2% in case of sexual violence [19]. This might be due to the cultural and traditional beliefs among family members that highlighted the importance of strong policy implementation [33] and providing sufficient protection from multiple and intersecting forms of discrimination for women and girls.

Even though the information collector was from the local community, some respondents may have underreported their experiences because the issue was sensitive to open discussion. In addition, the study was conducted in some schools among female students only, which was not able to address the issues of children out of school as well as the knowledge and perception of male students. So the findings may not represent the overall situation of children, which shows the need for a larger level study by including all the participants mentioned.

The Constitution of Nepal 2015 gives no woman the right to be subjected to physical, mental, sexual, psychological, or other forms of violence or exploitation [34]. Children are future adults, and it is necessary to prevent them from any type of violence because such experiences can have intergenerational effects. After school, students will start intimate relationships and will have their own families in the future, which can put them at greater risk of violence because major perpetrators of violence are intimate partners. So school students are required to be aware of the GBV, its effect on their lives, and preventive measures because the majority of them are not aware that they are the victims of GBV and it should be reported. As this study was a cross sectional study, the associations derived from it should be used prudently. In addition; study participants were only mid and late adolescents. So generalizing this result for whole adolescents should be done with caution.

## Conclusion

In conclusion, GBV, including physical, sexual, and emotional violence, is prevalent among female secondary school students. The drivers of physical violence were mainly family members, while sexual violence was from non-family members. Joint or extended types of family, discrimination between the genders within the family, living and educational status of parents, and experience of witnessing violence as a child were main factors associated with the GBV. This warrants urgent awareness in the community regarding female rights for their self-protection, protection from exposure to GBV since childhood and education on gender equality

within the family. Further, strong policy implementation to protect the girls from violence and proper reporting mechanisms through school and other appropriate channels are recommended.

## Supporting information

**S1 Data.**
(SAV)

**S1 File. Detailed questionnaire.**
(DOCX)

**S2 File. Inclusivity questionnaire.**
(DOCX)

**S3 File. Ethical approval.**
(DOCX)

## Acknowledgments

Special thanks to Education Officer of Brahmpuri Rural Runicipality Mr. Jay Prakash Shah, school principals, and teachers who supported during the data collection, as well as all the schools for their support. The support from all the members of the faculty of public health at the Manmohan Memorial Institute of Health Sciences is also appreciative. Above all, cordial thanks to all the respondents for their time, without whom this study would not have been possible.

## Author Contributions

**Conceptualization:** Laxmi Gautam, Manisha Shah, Durga Khadka Mishra.

**Data curation:** Manisha Shah.

**Formal analysis:** Laxmi Gautam, Manisha Shah, Sujan Gautam.

**Investigation:** Manisha Shah.

**Methodology:** Laxmi Gautam, Manisha Shah, Durga Khadka Mishra, Sujan Gautam.

**Project administration:** Manisha Shah.

**Resources:** Laxmi Gautam, Manisha Shah.

**Software:** Sujan Gautam.

**Supervision:** Laxmi Gautam, Sujan Gautam.

**Validation:** Laxmi Gautam, Durga Khadka Mishra, Padam Kanta Dahal, Sujan Gautam.

**Visualization:** Sujan Gautam.

**Writing – original draft:** Laxmi Gautam.

**Writing – review & editing:** Laxmi Gautam, Padam Kanta Dahal, Sujan Gautam.

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
