## [Decision Letter · Decision Letter 0]

21 Jun 2024

PGPH-D-24-01024

Prevalence and associated factors of gender-based violence for female: Evidence from school students in Nepal- a cross sectional study

Dear Dr. Gautam,

Thank you for submitting your manuscript to PLOS Global Public Health. After careful consideration, we feel that it has merit but does not fully meet PLOS Global Public Health’s publication criteria as it currently stands. Therefore, we invite you to submit a revised version of the manuscript that addresses the points raised during the review process.

EDITOR: We have completed the review now. Please provide point-to-point response to the reviewer's comments and concerns as a separate document.

We look forward to receiving your revised manuscript.

Kind regards,

Tanmay Bagade, Ph.D., MS (O&G), MPH, MHM

Academic Editor

Journal Requirements:

Additional Editor Comments (if provided):

Reviewers' comments:

Reviewer's Responses to Questions

**Comments to the Author**

1. Does this manuscript meet PLOS Global Public Health’s publication criteria? Is the manuscript technically sound, and do the data support the conclusions? The manuscript must describe methodologically and ethically rigorous research with conclusions that are appropriately drawn based on the data presented.

Reviewer #1: Yes

Reviewer #2: Yes

2. Has the statistical analysis been performed appropriately and rigorously?

Reviewer #1: Yes

Reviewer #2: Yes

3. Have the authors made all data underlying the findings in their manuscript fully available (please refer to the Data Availability Statement at the start of the manuscript PDF file)?

Reviewer #1: No

Reviewer #2: No

4. Is the manuscript presented in an intelligible fashion and written in standard English?

Reviewer #1: No

Reviewer #2: Yes

5. Review Comments to the Author

Reviewer #1: The manuscript needs to udergo a grammatical check to correct the several grammatical errors present throughout the document.

The results section needs to be re-done with the tables being made more presentable in terms of alignment of results with the variables as well as proper font size to ensure the content is easily readable. The text undder each table needs to decribe the results in the order that the they are presented in the table.

Throughout the document, the authors should ensure that each statement is complete and makes sense. Several statements are left hanging with no clear meaning.

With regards to data availabiilty, it is not clearly indicated where to access the data behind the finding presented in the manuscript.

Reviewer #2: This manuscript titled “Prevalence and associated factors of gender-based violence for female: Evidence from school students in Nepal- a cross sectional study” is a technically sound piece of scientific research with data that supports the conclusions. This cross-sectional study was conducted rigorously and the conclusions were drawn appropriately based on the data presented. Nevertheless, here are some observations for correction:

1. In the Abstract session, there is need to rephrase the first sentence of the conclusion.

2. Under introduction from 53 to 56 “GBV or violence against women and girls (VAWG), is a global pandemic that affects 1 in 3 women in their lifetime and 35% of women worldwide have experienced either physical and/or sexual intimate partner violence or non-partner sexual violence”. Kindly add references

3. Kindly delete (REF) in line 59

4. In line 60 “Studies in India(3) , Pakistan(4) and Maldives(5) revealed that 79.6%, 38.40% and 34.6% respectively” statement is not complete

5. It would be necessary to include the definition of gender based violence in the introduction

6. I suggest the need to include in your introduction the impact of gender based violence (on health, mental, social well-being) on female/women, and on the society at large

7. I recommend adding a stronger justification for conducting the study, for example what studies have been conducted in Nepal on GBV? What is known? What gap are you filling and why?

8. Under sampling design and population, more details are needed on the sampling technique “Two schools were selected using simple random sampling and probability proportionate allocation (PPA) was used to select the study population”. Was there any exclusion criteria? Kindly include

9. Can you add references to the sources of the question from these organizations “To measure these experiences different questions were developed based on the reference provided by WHO, UNICEF and UN women and were modified to make applicable in local context of the respondents”?

10. Did you explore getting informed consent from the parents of the minors? “Written informed consent was consent was taken from participants after explaining purpose of the study prior to the data collection. In case of participants below the age of 18 years, permission was taken from school principle, respective class teachers and then assent was taken from them”

11. The statistical analysis was prepared appropriately. However, my concerns were on the arrangement of the result. I suggest that descriptive statistics and tables come first before inferential statistics (bivariate and multivariate analysis). For the educational status of parents I suggest having a more standardized label such as Higher/Tertiary Education, Secondary Education, Primary Education and No formal education. Was there no mothers’ with higher education from the study? Why were there no values for fathers’ education following the multivariate analysis?

12. “To our knowledge, this is the first study to explore the prevalence and factor of GBV among the secondary school female students in Nepal”. Really?

13. “This study identified that the risk of GBV including physical, mental and sexual violence increasing among the secondary school girls where the joint family status, poor education status of parents, single parents and existence of alcohol or drugs abusers in the family.” Kindly rephrase

14. In discussing the prevalent of GBV is good to add implications of this findings

15. In conclusion, this manuscript assessed the prevalence and factor associated with gender-based violence among secondary school’s female students Nepal an implication for raising awareness, informing policy, challenging harmful gender norms, creating a safe environment for our girls and parental and community involvement. This research demonstrates a technically sound piece of scientific research with data that supports the conclusions. However, there is a need for copy editing to correct few typographical or grammatical at revision. I recommend the manuscript be accepted with minor revisions.

6. PLOS authors have the option to publish the peer review history of their article (what does this mean?). If published, this will include your full peer review and any attached files.

**Do you want your identity to be public for this peer review?** For information about this choice, including consent withdrawal, please see our Privacy Policy.

Reviewer #1: No

Reviewer #2: No

---

## [Editor Report · Decision Letter 1]

3 Sep 2024

Prevalence and associated factors of gender-based violence for female: Evidence from school students in Nepal- A cross-sectional study

PGPH-D-24-01024R1

Dear Ms Gautam,

We are pleased to inform you that your manuscript 'Prevalence and associated factors of gender-based violence for female: Evidence from school students in Nepal- A cross-sectional study' has been provisionally accepted for publication in PLOS Global Public Health.

Best regards,

Dr Tanmay Bagade

Academic Editor